# Drones and digital adherence monitoring for community-based tuberculosis control in remote Madagascar: A cost-effectiveness analysis

Lulua Bahrainwala[1], Astrid M. Knoblauch[2,3,4], Andry Andriamiadanarivo[5], Mohamed Mustafa Diab[6¤], Jesse McKinney[5,7], Peter M. Small[7], James G. Kahn[1,6], Elizabeth Fair[1,8], Niaina Rakotosamimanana[2], Simon Grandjean Lapierre[2,9,10]*

1 Institute for Global Health Sciences, University of California San Francisco, San Francisco, California, United States of America, 2 Institut Pasteur de Madagascar, Ambohitrakely, Antananarivo, Madagascar, 3 Swiss Tropical and Public Health Institute, Basel, Switzerland, 4 University of Basel, Basel, Switzerland, 5 ValBio Research Center, Ranomafana, Madagascar, 6 Philip R. Lee Institute for Health Policy Studies, University of California San Francisco, San Francisco, California, United States of America, 7 Global Health Institute, Stony Brook University, Stony Brook, New York, United States of America, 8 Department of Medicine, University of California San Francisco, San Francisco, California, United States of America, 9 Immunopathology Axis, Centre de Recherche du Centre Hospitalier de l'Université de Montréal, Montréal, Canada, 10 Microbiology, Infectious Diseases and Immunology Department, Université de Montréal, Montréal, Canada

☯ These authors contributed equally to this work.
¤ Current address: Global Health Institute, Duke University, Durham, North Carolina, United States of America
* simon.grandjean-lapierre.chum@ssss.gouv.qc.ca

**Data Availability Statement:** The primary data for this study is as described in the "Decision analysis model" and "Costs" sections and in "Table 1" and

## Abstract

### Background

Continuing tuberculosis control with current approaches is unlikely to reach the World Health Organization's objective to eliminate TB by 2035. Innovative interventions such as unmanned aerial vehicles (or drones) and digital adherence monitoring technologies have the potential to enhance patient-centric quality tuberculosis care and help challenged National Tuberculosis Programs leapfrog over the impediments of conventional Directly Observed Therapy (DOTS) implementation. A bundle of innovative interventions referred to for its delivery technology as the Drone Observed Therapy System (DrOTS) was implemented in remote Madagascar. Given the potentially increased cost these interventions represent for health systems, a cost-effectiveness analysis was indicated.

### Methods

A decision analysis model was created to calculate the incremental cost-effectiveness of the DrOTS strategy compared to DOTS, the standard of care, in a study population of 200,000 inhabitants in rural Madagascar with tuberculosis disease prevalence of 250/100,000. A mixed top-down and bottom-up costing approach was used to identify costs associated with both models, and net costs were calculated accounting for resulting TB treatment costs. Net

"Table 2". It includes Madagascar's National TB program data and TB disease pathophysiology information which is already in the public domain and appropriately referenced. It also includes. Madagascar's TB clinics operation cost and multiple bills and financial statements from the DrOTS project and implementing institutions. We believe the methods and detailed data sources provided in the manuscript text allows for reproduction of our results.

**Funding:** This work was supported by the Stop TB Partnership's TB REACH wave 5 award to NR and PMS (http://www.stoptb.org/global/awards/tbreach/wave5.asp). The Stop TB Partnership is funded by the Government of Canada and the Bill and Melinda Gates Foundation. AMK is supported by the Rudolf Geigy Foundation, Swiss Tropical and Public Health Institute, Basel, Switzerland. SGL is supported by the Canadian Association for Microbiology and Infectious Diseases. The funder did not provide salary support for principal investigators but did provide salary support for research staff including a co-author on this manuscript (AA). The specific roles of the authors are articulated in the 'author contributions' section." No funders or sponsors played any role in study design, data collection and analysis, decision to publish or preparation of the manuscript.

**Competing interests:** The authors have declared that no competing interests exist.

cost per disability-adjusted life years averted was calculated. Sensitivity analyses were performed for key input variables to identify main drivers of health and cost outcomes, and cost-effectiveness.

## Findings

Net cost per TB patient identified within DOTS and DrOTS were, respectively, $282 and $1,172. The incremental cost per additional TB patient diagnosed in DrOTS was $2,631 and the incremental cost-effectiveness ratio of DrOTS compared to DOTS was $177 per DALY averted. Analyses suggest that integrating drones with interventions ensuring highly sensitive laboratory testing and high treatment adherence optimizes cost-effectiveness.

## Conclusion

Innovative technology packages including drones, digital adherence monitoring technologies, and molecular diagnostics for TB case finding and retention within the cascade of care can be cost effective. Their integration with other interventions within health systems may further lower costs and support access to universal health coverage.

## Introduction

Tuberculosis (TB) is the leading infectious disease killer globally, surpassing HIV/AIDS and malaria, with 10 million new cases estimated in 2019 [1]. Among those, 3 million cases are classified as "missing", meaning they are undiagnosed or unreported and thus contributing to the continued epidemic. Innovative case finding, and control interventions are needed as conventional approaches alone are predicted to fail to meet the World Health Organization (WHO) End TB Strategy's objectives including that of eliminating TB by 2035 [2, 3]. In 2019, the Madagascar National TB Control Program (NTP) notified 34,191 TB cases. This represented only 56% of the country's annual incident cases as estimated by WHO [4].

Some of the biggest challenges low- and middle-income countries (LMIC) face for quality TB care include program underfunding, limited financial resources absorption capacity, low geographic coverage of centralized healthcare systems, paucity of trained healthcare personnel, weak diagnostic laboratory infrastructure, remoteness of rural communities, and limited transportation infrastructure that is sometimes exacerbated by difficult weather conditions [5–7]. To overcome some of these impediments, innovative technologies such as drones to facilitate clinical sample movement and strengthen healthcare supply chain were shown to have added value in specific settings [8–10]. Digital adherence monitoring technologies also represent a promising approach to improve and remotely monitor TB treatment adherence [11].

The use of drones in global health settings has rapidly evolved in the last years. As drone technology applications continue to expand beyond the military and recreational sectors, it is foreseen that drones will have a significant role to play for healthcare delivery in high- and low-income countries where either urgency, such as in the case of organ transplantation or cardiac arrest, or transportation complexity, such as in the case of last-mile delivery for enclaved population will justify their use [12, 13]. Drones also have the potential to revolutionize healthcare supply chain systems, facilitate surveillance in geographically challenging regions and support emergency response in face of life-threatening patient conditions or natural disasters [8, 14–16]. Despite the roll out of multiple drone pilot projects, especially in sub-

Saharan Africa, very little data is available on the health impact and costs of drone-supported interventions prior to this study [8].

In August 2017, the Drone Observed Therapy System (DrOTS) project was launched in Ifanadiana district, in southeastern Madagascar. DrOTS is a bundle of innovative interventions supporting community-based TB case finding including (i) drones to deliver sputum samples and TB medication between rural communities and diagnostics and treatment facilities; (ii) GeneXpert™ MTB/RIF (Cepheid, Sunnyvale CA USA) molecular platform to increase sensitivity and specificity of TB diagnosis and; (iii) WHO endorsed evriMED™ (Wisepill, Somerset West, South Africa) digital adherence monitoring technology to remotely assess TB treatment adherence by monitoring daily opening of an electronic pill box [17, 18]. In the context of isolated remote populations, this technology-supported model of TB care potentially allows for increased case finding and notification, faster diagnosis, and increased patient retention at every step of the cascade of care by favoring a patient-centric and community-based approach.

The global market for innovations like drones and evriMED™ devices in healthcare is rapidly developing and technology-associated costs are decreasing. Initially restricted to product developers and academia, the use of such technologies is now democratized to non-governmental organizations (NGO) and the public health sector allowing for larger scale-up [15, 19]. For NTPs in LMICs with limited financial and human resources, detailed economic evaluations of new programmatic strategies including such technologies are indicated to inform affordability and resource allocation priorities.

We present a cost-effectiveness analysis (CEA) of DrOTS compared to Directly Observed Therapy Short course (DOTS) from a healthcare system perspective [20]. DOTS is accepted as standard of care for TB treatment and adherence monitoring in most LMICs, including Madagascar. This CEA model sets a baseline for the economic evaluation of the use of drones and other innovative technologies for TB control.

## Materials and methods

In this CEA study, we use a decision analysis model to calculate the incremental cost-effectiveness of the DrOTS strategy compared to DOTS. This CEA study was conducted according to recommended methods and reporting for health care sector perspective CEA and a reporting checklist for cost-effectiveness analyses is provided as supplementary materials [21]. The overall DrOTS study, including the cost-effectiveness analysis, received ethical approval from both the Stony Brook University Internal Review Board (CORIHS# 2017-4056-F) and the "Comité d'Éthique de la Recherche Biomédicale" from the Ministry of Public Health in Madagascar (073-MSANP/CERBM).

### Study area and target population

Ifanadiana, a geographically remote district in the southeastern part of Madagascar, served as the use case typifying remote, impoverished and high TB burden areas. In this district, TB control activities are coordinated from a centralized diagnosis and treatment center (CDT) where a laboratory and pharmacy provide services to a population approximating 200,000 inhabitants. Twelve additional basic health care facilities in the district can supervise TB treatment administration, but initial diagnosis of TB is only performed in the CDT. The district is divided by a single paved road and access to the CDT from rural communities can require multiple days of walking which represents a significant barrier to access to care [22]. Madagascar's national TB guidelines recommend morning sputum smear microscopy on three consecutive days for initial TB diagnosis [23]. These can be prepared by nurses in the basic health care facilities and carried to the CDT by volunteer community health workers (CHW). In

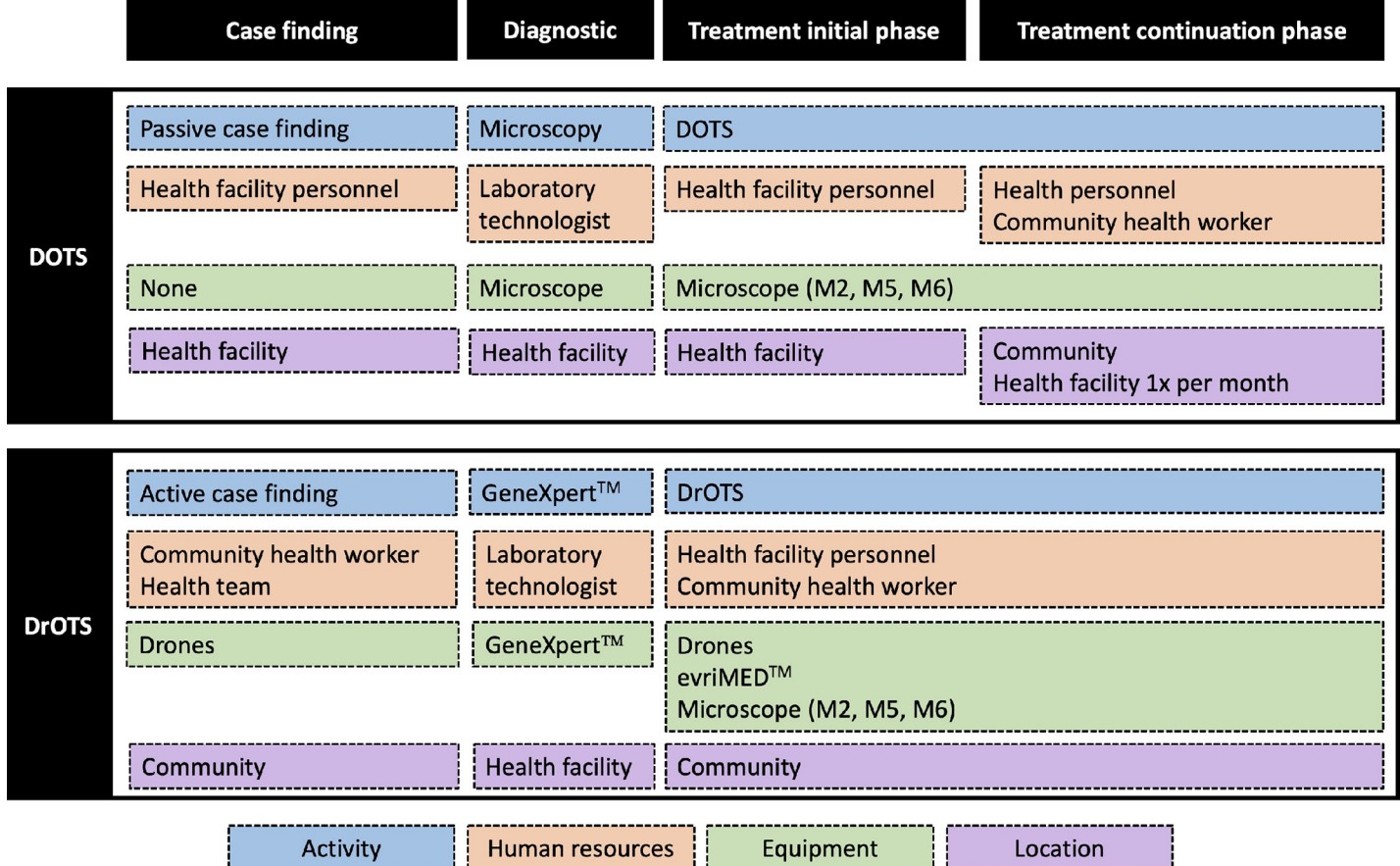

**Fig 1. Modeled TB care strategies.** DOTS; Directly Observed Therapy, DrOTS; Drone Observed Therapy System, GeneXpert™; GeneXpert™ MTB/RIF TB molecular testing platform, evriMED™; Digital adherence monitoring technology.

2018, 175 TB cases were reported in Ifanadiana, what corresponds to a notification rate of 88/100,000, less than one third of the county's average rate. The DOTS completion rate in the district was locally assessed to be 60% whilst the national successful treatment rate was reported to be 82% [24].

## Modeled strategies

The incremental cost-effectiveness ratio (ICER) measures the economic value of one intervention or strategy compared to another [25]. In our CEA, we compare conventional DOTS as per national TB guidelines, with DrOTS (Fig 1). For both models, primary field data was collected simultaneously in August 2017. For both modeled strategies, the total number of tested, diagnosed and treated individuals are presented in the results section.

**DOTS.** DOTS (Fig 1) relies on passive case finding, a strategy where symptomatic patients self-report to CHW or healthcare facilities. If appropriately tested and diagnosed with TB using smear microscopy, patients are initiated on therapy. DOTS requires patients to visit the treatment center on a daily basis for direct observation of medication intake during the initial 2 months of treatment. In cases of severe disease or incapacity to walk the necessary distance on a daily basis, patients should relocate close to the healthcare center during this period. Drug dispensing thereafter occurs monthly for the 4-months continuation phase of treatment

during which patients take medication back to their household. Adherence monitoring is based on pill count at the end of each month and therapy success is assessed through control sputum smear microscopy after 2, 5 and 6 months of treatment.

**DrOTS.** In DrOTS (Fig 1), active case finding, and innovative technologies are combined to facilitate community-based TB control. CHW perform bi-annual community outreach activities and identify symptomatic individuals and confirmed TB cases household contacts at village level. CHW then signal the health facility, via cell phone or global positioning system (GPS) tracker communication, and a drone travels from the CDT to the targeted remote community. One sputum sample from a person with suspected TB is flown back to the CDT where GeneXpert™ MTB/RIF testing is performed. In case of confirmed diagnosis, one month of anti-TB medication in an evriMED™ device is flown back to the patient. CHW equipped with tracker communication system assist this process and ensure all patients are provided with the appropriate medication at the right time. For modeling purposes, the impact of drones is captured by directly increasing the probability of TB suspects accessing TB diagnostic services and being tested for TB. Despite their presence in communities in DOTS, CHW don't have the ability to overcome the transportation challenges related to ensuring access to TB diagnostic services for all TB suspects. DrOTS, medication is dispensed this way on a monthly basis requiring a total of 7 round-trip flights for a full course of therapy. As with DOTS, sputum smear microscopy is used as a surrogate marker for treatment adherence and efficacy at months 2, 5 and 6. Activities are coordinated by the laboratory technician at the CDT and the drone technician and drones are stored and maintained at the CDT. An hybrid quad-copter and fixed-wing drone performs vertical landing and take-off at predetermined GPS coordinates allowing semi-automated bi-directional transport of 2.2 kg of cargo on a distance of 120 km [16]. Sample transport procedures adhere to WHO biosecurity guidelines for infectious substances transportation [26]. Flight trajectory is monitored by GPS-assisted piloting software which allows real-time location tracking. evriMED™ digital data are combined with remaining pill-count to monitor patient's adherence to therapy and adjust follow-up on a monthly basis.

## Modified strategies

To better assess the impact of specific innovations on cost-effectiveness, we modeled strategies where DOTS is augmented with individual innovations in a modular approach and where those same interventions are individually removed from the DrOTS bundle. We made explicit assumptions regarding how modifications affected diagnosis and treatment outcomes. This sub-analysis provides insights, though hypothetical, on the cost-effectiveness of using adherence monitoring technologies, GeneXpert™ MTB/RIF and drones independently for tuberculosis control in remote African populations.

## Decision analysis model

To calculate the ICER of DrOTS, a decision analysis model was designed using TreeAge Software (Tree Age Pro 2017, Health Care Edition; Williamstown, USA), and cross-checked against simplified calculations in R (R Foundation for Statistical Computing, Vienna, Austria). The model estimates costs, clinical outcomes, and disability-adjusted life years (DALYs) associated with DOTS, DrOTS and modified strategies over a 6 months period what corresponds to a complete treatment course for drug-susceptible pulmonary TB. A simplified representation of the decision tree is shown in Fig 2.

DrOTS and DOTS model assumptions are detailed and referenced in Table 1. When possible, assumptions were based on field data collected in Madagascar by authors. In absence of primary data, Madagascar public health data or published data was used to build the model.

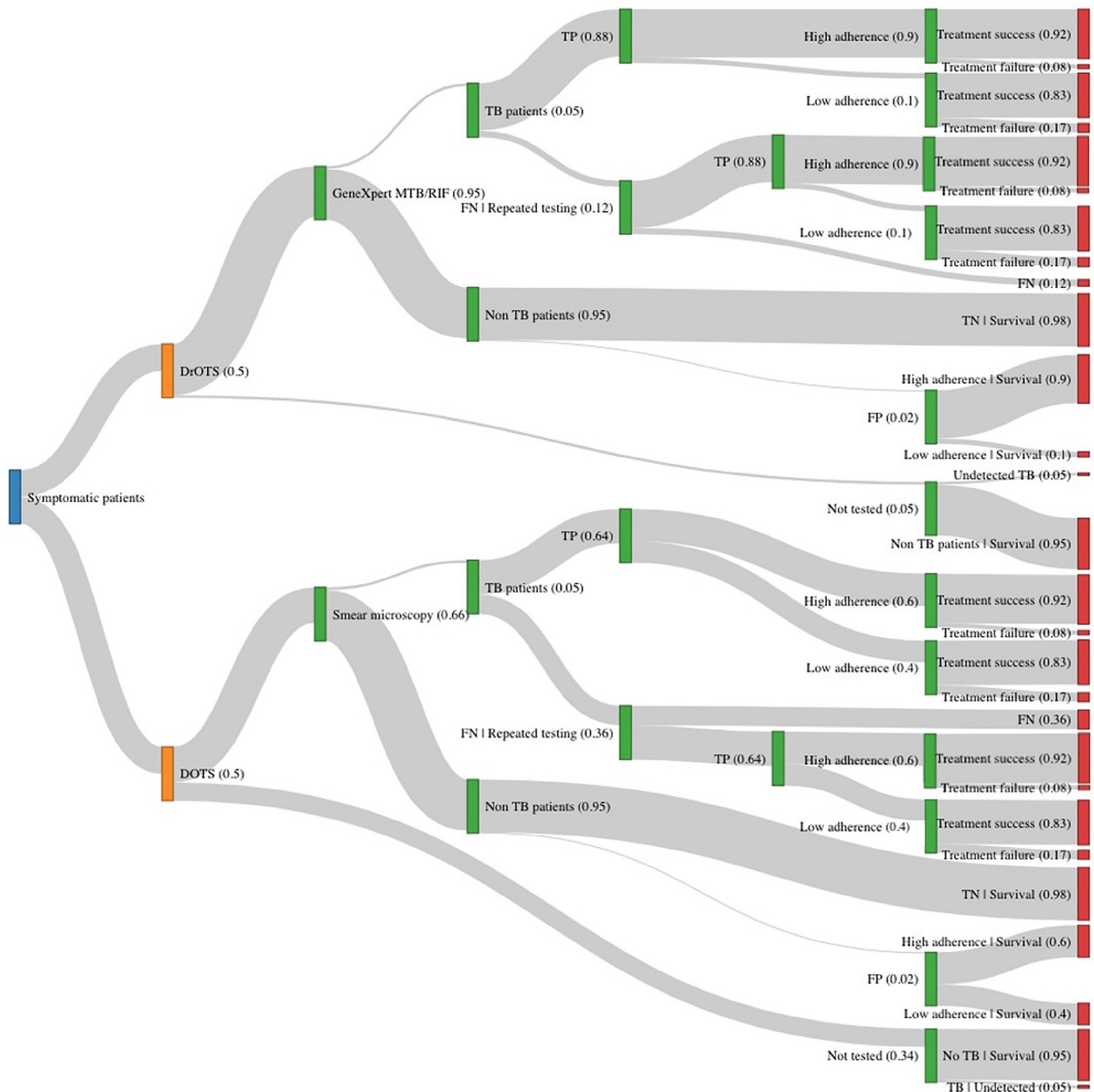

**Fig 2. Decision tree analysis model.** Decision tree analysis model comparing both strategies and presenting the study population (blue), TB control strategies (orange), decision nodes (green) and outcomes (red). Model arms width are proportional to probability scores described in text. DOTS; Directly Observed Therapy, DrOTS; Drone Observed Therapy System, FN; False negatives, FP; False positives, TN; True negatives, TB; Tuberculosis, TP; True positives.

To account for under-reporting to the NTP and improve external validity for high burden countries, prevalence of TB in the model population is fixed to 250/100,000 people [1]. According to limited published data on the prevalence of chronic respiratory symptoms in

**Table 1. Model inputs: Parameters, model assumptions, and sensitivity analyses ranges.**

| Parameter | | Model assumptions | Range | Source, reference |
|---|---|---|---|---|
| | TB prevalence in study population (per 100,000) | 250 | 100–1,000 | [1, 4, 28] |
| | Chronic respiratory symptoms prevalence (%) | 5 | 1–25 | [27] |
| | Willingness to produce sputum for laboratory testing (%) | 95 | NA | [5] |
| | Laboratory testing of submitted clinical samples (%) | 100 | NA | Model assumption |
| | Enrolment into care following TB diagnosis (%) | 100 | NA | Model assumption |
| | Probability of treatment success, with high adherence (%) | 92 | NA | [29] |
| | Probability of treatment success with low adherence (%) | 83 | NA | [29] |
| DOTS | | | | |
| | Probability of being tested for TB—Passive case finding (%) | 66 | 30–90 | [4, 28, 30, 31] |
| | Sensitivity of sputum smear exam (%) | 64 | NA | [32] |
| | Specificity of sputum smear exam (%) | 98 | NA | [32] |
| | Probability of high treatment adherence (%) | 60 | 40–95 | Field collected data |
| DrOTS | | | | |
| | Probability of being tested for TB—Active case finding (%) | 95 | 50–99 | Model assumption |
| | Sensitivity of GeneXpert™ MTB/RIF (%) | 88 | NA | [17, 32] |
| | Specificity of GeneXpert™ MTB/RIF (%) | 98 | NA | [17, 32] |
| | Probability of high treatment adherence (%) | 90 | 50–99 | [11, 33] |
| Age, life expectancy, DALYs and discount rate | | | | |
| | Average age of diagnosis (years) | 25 | NA | [34] |
| | Age of early death due to TB (years) | 28 | NA | [34] |
| | Pre-diagnosis active pulmonary disease duration (months) | 6 | NA | [34] |
| | Treatment duration (months) | 6 | NA | [23, 32] |
| | Life expectancy at birth (years) | 63.6 | NA | [34, 35] |
| | Disability weight due to TB | 0.333 | NA | [35] |
| | Discounting rate (%) | 5 | NA | [36] |

DALYs, disability adjusted life years; DOTS, directly observed therapy short course; DrOTS, drone observed therapy system; MTB, *Mycobacterium tuberculosis*; NA, not applicable; RIF, rifampicin; TB, tuberculosis.

similar Malagasy districts, we estimate that minimally 5% of the screened subjects will present at least one TB symptom and thus be eligible for sputum testing [27]. We further assume that 95% of symptomatic patients will provide sputum for laboratory testing in both care models [5]. For modelling purposes, all submitted samples are assumed appropriately tested and all patients are assumed appropriately started on treatment if found to be infected with TB in both strategies tree arms.

Confronting available baseline demographic data and reported district TB data to estimated WHO country incidence suggests that conventional DOTS achieves screening of two-thirds of symptomatic patients through passive case finding [4, 28, 30, 31]. We assume that 95% of individuals are screened during DrOTS active case finding systematic community outreach activities. Sputum smear microscopy and GeneXpert™ MTB/RIF assays sensitivity and specificity characteristics for diagnosis of active pulmonary disease are obtained from the literature [17, 32]. Active case finding generally results in increased case detection and decreased positivity rates because of broader testing [37]. Diagnostic assay positivity rates and tests predictive values are not included in the here presented model.

Drug adherence below the 95% and 90% thresholds is, respectively, associated with at least 8% and 17% of treatment failure [29]. We therefore used the 90% total adherence threshold to categorically differentiate high adherence (>90% total adherence) and low adherence (<90%

total adherence) patients. Treatment outcomes were identically modeled according to those thresholds in both arms. Based on primary data collected in TB clinics in the study area, the probability of being highly adherent under DOTS was found to be 60%. This calculation was based on surrogate adherence markers which include providers' reporting, NTP recorded treatment success rate, and the matching of prescribed doses with the anticipated number of doses included in a standard 6 months regimen [28]. In the DrOTS arm, where treatment adherence is supported and measured by evriMED™ devices, no data from Madagascar or other rural sub-Saharan Africa is yet available. Probability of patients being highly adherent is estimated to be 90% based on a systematic review on digital adherence monitoring technology and a large-scale trial in rural Chinese communities [11, 33, 38].

Both DOTS and DrOTS patients testing falsely negative on smear microscopy or GeneXpert™ MTB/RIF are expected to self-present to care a second time because of ongoing symptoms and be re-tested for TB. TB patients either refusing testing or not being screened go untreated. For those testing falsely positive, TB treatment is initiated, and costs associated with TB are included. Costs associated with missed alternative diagnosis, negative health outcomes and potential medication undesired effects are not considered in the analysis. This model focuses on prevalent cases but does not take into account incident cases in the study villages which would occur during the 6 months span or incident cases which would be prevented because of index cases diagnosis and management. Subjects under 5 years of age are excluded from the analysis acknowledging the fact that childhood TB diagnosis and management requires a personalized approach in closer collaboration with centralized healthcare facilities.

## Disability adjusted life years

DALYs were chosen as outcome measure of both modelled TB care strategies to ensure comparability between TB control strategies [39, 40]. Life expectancy, disease duration and disability weights and present-value discounting were considered in the DALYs estimates. Once patients successfully complete treatment, it is assumed that they subsequently live to full life expectancy as is the case for those unaffected by disease. Based on published case series and country-specific data in Madagascar, the average age at diagnosis is estimated at 25 years, total life expectancy is 63,6 years and the weight index used to calculate pre-fatal health loss attributable to time lived with active TB disease is 0.333 (95% CI: 0.224–0.454) as suggested by the Global Burden of Disease 2013 study [34, 35]. The duration of symptomatic active pulmonary disease leading to diagnosis is 6 months and patients going undiagnosed or failing therapy are assumed to succumb within 3 years [34]. Drug susceptible TB treatment is included in duration of disease and is assumed to be limited to 6 months. Using the lifetime horizon, a 5% discount rate and these inputs, we calculated DALYs expected due to diagnosis and treatment of TB and DALYs expected due to untreated TB for both TB care strategies [36, 41].

## Costs

Total costs are calculated in USD per TB patient and inflated to 2019 with an annual inflation rate of 4.9% for expenses and costing data predating the cost analysis [42]. Using field collected and literature data, a mixed micro and macro costing approach was used to identify costs associated with both care models. Common and strategy-specific elements of costs for DOTS and DrOTS are detailed and referenced in Table 2.

Costs of TB medication, NTP indirect expenses and patient support are the same for both care models, were estimated based on the WHO's 'CHOICE–Unit costs estimate for service delivery tool' and were cross validated with NTP officials and TB clinic managers in Madagascar [43]. NTP indirect costs include salaries of program managers, coordination and

**Table 2. DOTS and DrOTS elements of costs.**

| Cost element | Cost (USD) | Source, reference |
|---|---|---|
| Drugs for treatment of drug-susceptible TB | 39.79 | [43] |
| Patient support costs | 41.89 | [43] |
| NTP indirect program costs | 14.66 | [43] |
| NTP human resources costs | 14.66 | [43] |
| Unit cost sputum smear microscopy | 2.25 | [43, 44] |
| Untreated TB costs | 209.00 | [45] |
| Madagascar GDP | 460.73 | [46, 47] |
| WHO willingness to pay threshold | 1,382.19 | [39, 46] |
| DOTS | | |
| Cost per outpatient visit | 0.95 | [43] |
| DOTS human resources | 0.52 | Field collected data |
| DrOTS | | |
| Drone system start-up cost annualized over a 5-year time period | See Table 3 | Field collected data, [8, 18, 44] |
| Drone specific maintenance cost | See Table 3 | Field collected data |
| Cost per round trip drone flight | 1.52 | Field collected data |
| GeneXpert MTB/RIF acquisition annualized over a 5-year time period | 6,039.26 | Field collected data |
| Unit cost GeneXpert™ MTB/RIF testing | 15.87 | [44] |
| evriMED™ device acquisition annualized over a 5-year time period | 3,900.41 | Field collected data |
| DrOTS human resources | 11.65 | Field collected data |
| Cost per interaction with CHW | 0.50 | Field collected data |

CHW, Community health worker; DOTS, directly observed therapy short course; DrOTS, drone observed therapy system; GDP, gross domestic product; MTB, *Mycobacterium tuberculosis*; NTP; national tuberculosis control programme; RIF, rifampicin; TB, tuberculosis; USD, United States dollars.

equipment at national and regional levels. Patient support includes nutritional aid, transportation vouchers for medical visits and other in-kind benefits given to TB patients as recommended by national TB guidelines [23]. Costs related to untreated or unsuccessfully treated TB in Madagascar were estimated using LMIC-specific costing data from a systematic review [45]. Those costs include repeated medical consultations or hospitalizations, therapeutic interventions and adverse events management associated with incorrectly diagnosed non-TB respiratory conditions and potential need for laboratory screening of multidrug-resistant TB (MDR-TB) due to repeated inadequate management. Unit cost of sputum smear microscopy used for initial diagnosis in DOTS and for per-treatment control testing in both care models is based on literature and WHO assessment [43, 44]. For patients initially testing falsely negative, costs associated with laboratory diagnosis are doubled in the cost analysis accounting for repeated consultation with persistent symptomatology.

DOTS specific costs include those associated with outpatient visit, facility-based human resources and laboratory microscopy initial diagnosis. The operation costs of district (CDT) and lower levels facilities involved in TB patient care are used to calculate outpatient visits costs. Human resources include salaries of physician, laboratory technician and nursing personnel adjusted to the number of anticipated interactions a TB patient has with these respective providers over one TB episode treatment course. Costs associated with the necessity for patients to relocate near treatment center for the 2-months initial treatment phase are not

**Table 3. Detailed costs associated with start-up and maintenance of DrOTS.**

| Component | | Cost (USD) | | Source |
|---|---|---|---|---|
| | | Total | Annualized | |
| **Start-up costs** | | | | |
| **Technology procurement** | | | | |
| | Drones | 237,539 | 54,869 | Field collected data |
| | Batteries | 27,229 | 6,290 | Field collected data |
| | Power supply and charger | 4,780 | 1,104 | Field collected data |
| | Computers | 83,333 | 19,250 | Field collected data |
| | Information technologies and communications | 26,118 | 6033 | Field collected data |
| | Packaging and shipment | 7,170 | 1,656 | Field collected data |
| | Taxes and customs | 356,296 | 82,304 | Field collected data |
| | Biosafety sample transport material | 28,200 | 6,514 | Field collected data |
| **Training** | | | | |
| | CHW | 33,391 | 7,713 | Field collected data |
| | NTP personnel | 27,838 | 6,430 | Field collected data |
| | Specialized drone personnel | 42,910 | 9,912 | Field collected data |
| | Server acquisition | 74783 | 17275 | Field collected data |
| | Flight software development and maintenance | 61772 | 14269 | Field collected data |
| | Mobile phones | 50,000 | 11,550 | Field collected data |
| **Maintenance costs** | | | Yearly | |
| | Internet connection | NA | 5,612 | Field collected data |
| | Server maintenance | NA | 2,309 | Field collected data |
| | Database maintenance | NA | 1,251 | Field collected data |
| | Insurance | NA | 26,967 | Field collected data |
| | Drones maintenance & storage | NA | 36,141 | Field collected data |

CHW, Community health worker; NTP, National Tuberculosis Control Program

included in the analysis since these are assumed by patients and families and this CEA is performed from a healthcare system perspective.

In DrOTS, start-up costs are associated with acquisition of innovative technologies including bi-directional drone delivery systems, GeneXpert™ MTB/RIF molecular diagnostics machine, evriMED™ adherence monitoring devices, and the necessary associated communication technologies and data management systems. Beyond procurement, start-up costs also include initial personnel training. Start-up costs are annualized over a five-year period as recommended by WHO guidelines [39]. Table 3 further breaks down the start-up and maintenance costs specifically associated with the bi-directional drone transportation program. These budgetary data were collected from project-specific programmatic planning documents and reports, purchase records, and tax receipts for drones and other supplies. Drone acquisition, batteries, power supplies and chargers are those paid by Madagascar project implementers at onset in 2017. Costs associated with computers include computers for all involved parties including technology experts, program management and field teams. Biosafety sample transportation materials represent mandatory UN3373 packaging for clinical samples transportation [26]. Packaging, shipment, taxes and customs fees are applied to all necessary materials unavailable in Madagascar. Other DrOTS specific costs include costs per outpatient diagnosis and per-treatment interactions with the CHW and additional human resources supporting field active case finding and drone operations.

For routine operations, specialized personnel are trained for the operation and maintenance of drones, whereas TB nurses and CHW are trained to safely ensure drugs or medical

supplies and clinical samples are respectively extracted and loaded in the drones. For involved NTP personnel and drone technicians, training occurs upon program launching and after 6 and 12 months of operations. At village level, CHW get intermittent training courses on TB and appropriate drone usage every four months. Fixed annual operation costs of the drone program include vehicle maintenance, flight insurance and storage. For cost associated with drone flight, we only included those associated with electricity for battery charging. Although flight purposes can be combined in the case were multiple TB patients are cared for in the same village, this potential economy of scale is inconsistent and thus not considered in this analysis. To ensure optimal communication, capacity to request drone transportation and coordination of return flights, CHW are provided with phones. In DrOTS, the cost of GeneXpert™ MTB/RIF cartridge are those associated with initial TB diagnosis.

Patients diagnosed with TB in both care models are systematically tested for HIV according to national TB guidelines [23]. Since prevalence of HIV-TB co-infection in Madagascar is less than 1%, costs related to HIV testing and management are not accounted for [4, 48]. Similarly, costs associated with hospitalization, longer treatment course and drug adverse events management in MDR-TB treatment are excluded since this represents less than 1% of cases in Madagascar and has never been diagnosed in the study area [4].

According to the World Bank and WolrdData.info, which compiles composite data from the World Bank and the International Monetary Fund, Madagascar's annual gross domestic product (GDP) per capita and average annual income were respectively $460.73 USD and $510 USD in 2018 [46, 49]. Multiple cost-effectiveness thresholds have been proposed to assess and prioritize interventions in the health sector. For this analysis, ICER values of three times Madagascar's per capita GDP ($1382.19), and one time per capita GDP ($460.73) were considered cost-effective, and very cost-effective, respectively [39, 50].

DALY, costs and the incremental cost of DrOTS over DOTS is calculated as follows:

$$DALY = (Life\ expectancy\ at\ birth - Age\ at\ death) + ((Age\ at\ death \\ - Age\ at\ diagnosis\ of\ disease)\ X\ (Disability\ weight))$$

$$\Delta\ DALY\ averted = (DALYs\ expected\ with\ DOTS) \\ - (DALYs\ expected\ with\ DrOTS),\ taking\ into\ account\ differences\ in\ diagnostic\ and\ treatment\ success.$$

$$\Delta Costs = (Net\ cost\ of\ DrOTS) - (Net\ cost\ of\ DOTS)$$

$$ICER = \Delta\ Costs/\Delta\ DALY\ averted$$

## Modified strategies

To further assess the specific impact of digital adherence monitoring technology, laboratory diagnostics performance and aerial transportation systems on cost-effectiveness of DrOTS, we built modified strategies models. Specifically, we assessed the cost and impact of evriMED™ devices for adherence, GeneXpert MTB/RIF for diagnosis accuracy, and drones to support CHW community case finding by adding each to the DOTS tree in turn. Similarly, we assessed the specific contribution of those interventions to the cost and impact of DrOTS by removing each from the DrOTS tree in turn.

## Sensitivity analyses

To assess the impact of uncertainty in the value of key input variables, one-way sensitivity analyses were performed across plausible ranges (Table 1). TB prevalence, prevalence of TB-compatible symptoms in target communities, probability of being tested in both care models,

probability of being highly adherent to therapy in both care models and DrOTS start-up costs were the inputs for which sensitivity analyses were performed. For a subset of those inputs, values beyond Madagascar's plausibility ranges were analyzed to increase external validity. For example, TB prevalence sensitivity analysis was performed between 100 and 1,000 cases per 100,000 to include Madagascar reported notification rates and that of other lower or high burden countries [1, 4].

## Results

### Program service quantity

The DOTS intervention strategy reached two-thirds of the population presenting chronic respiratory symptoms and 95%of those accepted to produce a sample for testing using smear microscopy. DOTS implementation in the study populations hence led to TB testing for 6,333 presumptive TB patients. Of these, 276 were confirmed to have active TB disease and enrolled into care (Table 4).

The DrOTS intervention strategy reached 95% of the symptomatic patients for GeneXpert MTB/RIF testing, with 9,025 patients being tested for and 445 being diagnosed and enrolled into care. This represents a 61.2% (95% CI 58.1–65.2, $p<0.05$) increase in case finding and treatment initiation over DOTS.

DOTS led to 244 (88.4%) successfully treated cases and 32 (11.6%) treatment failure cases. With the implementation of digital adherence monitoring technologies, those outcomes were respectively 405 (91.0%) and 40 (9.0%) in DrOTS. This represents a 2.6% (95% CI -1.8–7.5, $p = 0.47$) increase in successful outcomes.

Of the 500 prevalent TB cases in the modeled 200,000 population, DOTS and DrOTS, led to 256 (51.3%) and 95 (19.0%) TB-related deaths, respectively ($p<0.05$). The decision analysis tree model indicates that most of this difference was due to the different case-finding and treatment initiation strategies in DrOTS.

In absence of any intervention in the study population, TB was calculated to account for 7,949 DALYs. Considering an average age of 25 years at disease onset, a life expectancy of 63.6 years and a disability weight of 0.333, DOTS and DrOTS were respectively associated with a total of 4,157 and 1,643 DALYs at population level (Table 4). This corresponds to an average 8.3 and 3.3 DALYs per TB patient in the population and an incremental effect of 2,514 DALYs averted (4,157 DALYs for DOTS minus 1,643 DALYs for DrOTS) ($p <0.05$).

The net costs associated with DOTS and DrOTS implementation in the 200,000 population were respectively calculated to be $141,132 and $586,101 (Table 4). Net costs associated with initial case finding, diagnosis and clinical management per TB patient were calculated for both care models. The costs common to DOTS and DrOTS were $111. DOTS-specific additional costs were $29 for a total combined cost per TB patient of $140. DrOTS specific costs per patients were $837 for a total combined cost per TB patient of $948. When accounting for costs associated with undiagnosed TB and TB testing of symptomatic non-TB patients, DOTS

**Table 4. Projected outcomes and cost-effectiveness of DOTS and DrOTS for TB control in a 200,000-population remote district of Madagascar.**

| Strategy | Diagnosed TB cases | Deaths due to active TB | DALYS averted | Net cost ($USD) | Cost per TB patient ($USD) | Incremental cost per additional active TB case diagnosed | ICER |
|---|---|---|---|---|---|---|---|
| DOTS | 276 | 256 | 3,792 | 141,132 | 282 | NA | NA |
| DrOTS | 445 | 95 | 6,306 | 586,101 | 1,172 | 2,631 | 177 |

DOTS, directly observed therapy short course; DrOTS, drone observed therapy system; ICER, Incremental cost effectiveness ratio; TB, tuberculosis.

had a cost of $282 per TB patient in the total population. This number increased to $1,172 in DrOTS. The incremental cost per additional TB patient diagnosed in DrOTS was 2,631$.

DrOTS has an ICER value of $177 per DALY averted compared to DOTS for diagnosis and treatment of TB. This represents the incremental cost of DrOTS over DOTS (i.e. DrOTS intervention net cost of $586,101 minus DOTS intervention net cost of $141,132) divided by the difference in DALYs averted between both strategies (i.e. 6,306 DALYs averted in DrOTS minus 3,792 DALYs averted in DOTS).

## Drones, evriMED™ and GeneXpert™ MTB/RIF modified strategies models

Case finding, treatment outcomes, costs and DALYs were further compared between DOTS, DrOTS and six additional distinct modified strategies integrating various components of the DrOTS bundle. evriMED™, GeneXpert™ MTB/RIF and drones were differentially added to the standard of care DOTS model to assess the individual impact of those interventions on disease control and cost-effectiveness. Similarly, those interventions were differentially removed from the DrOTS model to assess their individual contribution to the bundle impact and cost-effectiveness. Table 5 presents outcomes for all strategies in cost-effectiveness incremental order.

Given that drones allow for sample transportation and facilitated access to diagnosis facility, their impact on the decision tree is represented by increased probability of being tested for TB. Although CHW are present in villages within DOTS, the model assumption is that this alone doesn't correlate to higher probabilities of being tested for TB given the transportation impediments to access to diagnostic laboratories. Adding drone-supported outreach case finding to DOTS yielded an additional 2,692 (41.6% increase) patients for TB testing and 117 (42.3% increase) TB case notifications. Even without altering diagnostic testing sensitivity with GeneXpert™ MTB/RIF implementation, drone-supported outreach activities led to an increase in case finding. The ICER of this strategy over conventional DOTS was measured to be 222 and the net cost per additional TB patient diagnosed was $2,631. Given the increased sensitivity of GeneXpert™ MTB/RIF for diagnosis of active TB disease, replacing smear microscopy by molecular diagnostic in DOTS led to an additional 36 (13.0%) TB patients diagnosed for an ICER of $115 over DOTS. Purely increasing treatment adherence with evriMED™ in DOTS led to 7 (2.9%) additional patients successfully completing therapy and averted TB-related deaths for an ICER of $33. Among all three innovative strategies, increasing case finding using drones had the highest related number of DALYs averted but came at the highest price.

Removing drone-supported CHW outreach case finding from DrOTS led to a reduction of 133 (29.9%) TB notifications and a 121 (56.0%) increase in TB-related deaths. The ICER of DrOTS decreased from $177 to $94 by doing so. Drones were found to be the technology most significantly contributing to DrOTS net costs and ICER. Linking DrOTS to conventional microscopy by removing GeneXpert™ MTB/RIF from the intervention bundle led to 52 (11.7%) fewer TB notifications and increased the intervention ICER from $177 to $193. Similarly Linking DrOTS to conventional adherence monitoring led to 8 (2.0%) fewer patients successfully completing therapy and averted TB-related deaths for an ICER increase from $177 to $190. Both "DrOTS without GeneXpert™ MTB/RIF" and "DrOTS without evriMED™" scenarios showed that bridging DrOTS to conventional diagnostics or digital adherence monitoring technology came at a lower net cost but had a higher ICER.

As presented in Table 5, incrementally adding interventions to strategy bundles did not always correlate to increased cumulative costs per TB patients diagnosed and ICER. For example, implementing DrOTS without GeneXpert MTB/RIF™ represented an economy of $197 per additional TB patients diagnosed and still lead to an increase in ICER of $3 over the

**Table 5. Cost-effectiveness of DOTS, DrOTS and modified strategies for TB control in a 200,000-population remote district of Madagascar.**

| Strategy | Diagnosed TB cases | Deaths due to active TB | DALYS averted | Net cost ($USD) | Cost per TB patient ($USD) | ICER over DOTS ($USD / DALY averted) | Cumulative incremental cost per additional active TB case diagnosed (USD) | Cumulative ICER ($USD / DALY averted) |
|---|---|---|---|---|---|---|---|---|
| DOTS | 276 | 256 | 3,792 | 141,132 | 282 | N/A | N/A | N/A |
| DOTS with evriMED™ | 276 | 249 | 3,908 | 145,032 | 290 | 34 | 8 | 34 |
| DrOTS without drones | 312 | 216 | 4,425 | 200,545 | 401 | 94 | 111 | 60 |
| DOTS with GeneXpert MTB/RIF™ | 312 | 224 | 4,294 | 198,751 | 398 | 115 | -4 | 21 |
| DrOTS | 445 | 95 | 6,306 | 586,101 | 1,172 | 177 | 775 | 62 |
| DrOTS without evriMED™ | 445 | 107 | 6,120 | 582,201 | 1,164 | 190 | -8 | 13 |
| DrOTS without GeneXpert MTB/RIF™ | 393 | 142 | 5,569 | 483,943 | 968 | 193 | -197 | 3 |
| DOTS with Drones | 393 | 153 | 5,404 | 499,323 | 999 | 222 | 31 | 29 |

Strategy bundles outcomes for all in cost-effectiveness incremental order. DOTS, directly observed therapy short course; DrOTS, drone observed therapy system; ICER, Incremental cost effectiveness ratio; TB, tuberculosis

previous best strategy (DrOTS without everiMED™). Implementing DOTS with drones proved to be the strategy with the most unfavorable ICER at $222.

## Sensitivity analyses and cost-effectiveness thresholds

Sensitivity analyses were performed around model input variables with a higher level of uncertainty (Fig 3). Decreasing TB treatment adherence was shown to have a significant detrimental effect on DrOTS ICER with costs per averted DALYs reaching $536.94 when treatment high adherence rate is reduced to 50%. DrOTS ICER was also show to be heavily dependent on case finding effectiveness and probability of being tested with the ICER increasing to 497.75 USD per DALY averted when the probability of being tested within DrOTS dropped to 70%. The ICER of DrOTS over DOTS ranged between 49.95 and 443.44 USD per DALY averted for TB prevalence rates in the populations respectively varying between 100 and 1,000 per 100,000 population. DrOTS initial costs had a smaller effect than other factors on the ICER between half ($530,677.92) and twice ($2,122,711.66) the costs associated with the base case scenario, DrOTS ICER varied from 131.21 and 280.90 USD per DALY averted.

Using model assumption values, DrOTS and all modified strategies' ICER compared to DOTS were lower than one annual GDP per capita willingness to pay threshold of $460.73. For all selected input variable plausibility ranges included in the sensitivity analysis, the ICER also remained below this threshold. To reach this health system willingness to pay threshold, TB prevalence had drop to 96/100,000 population what represents a 58.8% decrease from Madagascar's current rate. Similarly, the prevalence of TB compatible symptoms in the study population had to increase to 38.5% for DrOTS to cross the willingness to pay threshold because of costs associated with testing of non-TB patients. Only by increasing the initial costs associated to DROTS by 3.8 times to a total of 4,033,152$ did the model confirm an ICER of 460.73 for DrOTS. When assessing specific DOTS-related input variables, the probability of being tested had to increase to 93.4% for DrOTS to reach the 460.73 ICER threshold. Even

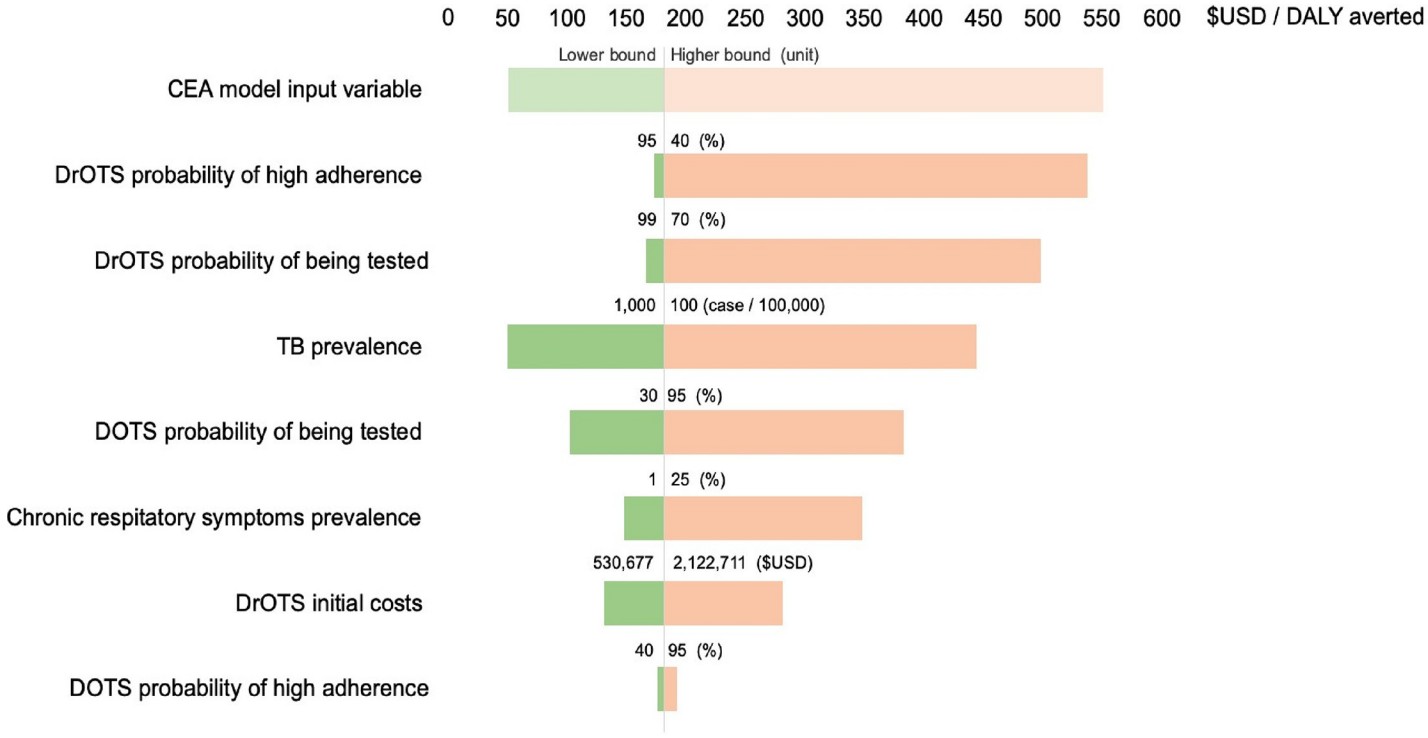

**Fig 3. Incremental cost-effectiveness ratio of DrOTS over DOTS–sensitivity analysis.** Sensitivity analysis around key input variables of DrOTS versus DOTS incremental cost effectiveness ratio analysis.

with a probability of being highly adherent of 100% within DOTS, the ICER of DrOTS was still 193.26. For DrOTS, the probability of being tested and of being highly adherent to therapy respectively had to drop to 71.1% and 52.2% for the intervention to reach the threshold ICER.

## Discussion

In this study, we leveraged primary costing, operational and impact data from the DrOTS program in remote Madagascar to build a modeled CEA comparing this novel approach to DOTS standard of care. We show that a bundle of innovative interventions for TB control including drones to transport sputum samples and medication, evriMED™ digital adherence monitoring technology and GeneXpert™ MTB/RIF is cost effective at an ICER of 177 USD per DALY averted. This is less than the annual GDP per capita which is proposed as a very cost-effective threshold.

The modeled CEA suggests that linking drones to improved laboratory diagnostics and digital adherence monitoring technology is essential to further increase cost-effectiveness. Similarly, implementing drones within a standard of care DOTS program was shown to improve disease control but at a higher ICER than any other approach (ICER = $222 USD per DALY averted).

Sensitivity analysis further shows that at the low willingness to pay threshold of Madagascar's one GDP per capita ($ 460.73 USD) can still be met in lower TB prevalence settings and for higher drone program initial acquisition costs. This suggests favorable cost-effectiveness ratios in other high burden countries with different disease epidemiology and economic realities. To our knowledge, using TB control in remote Madagascar, this study is the first study to

present a cost effectiveness evaluation of bi-directional drone transportation in the health care sector. It provides reliable and timely data which donors, governments and in-country implementors should consider in order to guide programs design and resources allocation. It also sets a benchmark for future programs drone technology acquisition net costs and intervention related averted DALYs. Our study design permits comparison of the impact on standardized TB indicators, the costs and ICER of drones with other diagnostic and adherence increasing interventions with which the TB community is familiar and for which a more significant burden of medical and economic literature is available.

The market for drone technology is currently evolving at unprecedented speed. Our costing analysis is based on technology initial acquisition costs for the DrOTS project and represents a current best estimate. Prices and discounting rates for multiple elements of the DrOTS intervention strategy will rapidly evolve in time as the market for drones in global health further takes shape and technologies are deployed at scale.

It should be kept in mind when analysing the presented data that, given the healthcare system perspective of this analysis, patients costs and loss of opportunity as well as societal costs associated with TB disease are not measured. In Madagascar, most of the population lives in remote villages and survive on subsistence farming. TB incurs catastrophic health expenditures, productivity losses and opportunity costs, for patients. Those should be further explored to complement this CEA and provide a societal cost perspective. Determining the success of innovative technologies implementation in healthcare cannot be limited to the evaluation of their impact on standardized metrics and costs. Though such measures are central to establishing their added value, the success of innovative public health interventions is also contingent on how those on the receiving end perceive their roll-out. Hence, formal acceptability and cultural perceptions studies should also be performed [51].

This CEA model has several limitations. Our study focuses on case finding and management of active pulmonary TB disease. The impact of latent infection, incipient TB, or extrapulmonary forms of active disease was not measured. Those forms of TB infection either represent a reservoir or an unmeasured burden of DALYs which were not part of the analysis. Our model focused on prevalent cases on a six-month intervention period. It did not account for retreatment, failures and relapses or for the long-term impact on incidence of reducing the burden of active disease. Although their prevalence is low in Madagascar, the absence of consideration for HIV-coinfections and MDR-TB infections may limit the external validity of our study for countries where they represent significant public health challenges. As any model, this study's CEA tends to simplify many aspects of TB care. For all evaluated strategies, we assumed systematic testing of submitted samples and enrollment into care for all diagnosed cases. Those two potential gaps in the TB cascade of care have many determinants but it is assumed that those gaps would be wider in the DOTS model therefore favourably increasing the perceived effectiveness of DrOTS over DOTS. Isolating the net effect of drones in a comprehensive care model represents another challenge. In this analysis, drones were modeled as a mean to improve access to TB diagnostics for TB suspects by overcoming the transportation challenges leading to significant numbers of recognized but untested patients or lost or samples in remote Madagascar. Their indirect effects such as improved adherence because of faster medication availability or diagnosis performance because of better conservation and shorter transit delays of laboratory samples were not calculated but would most likely increase their perceived added value and ICER.

Significant transportation infrastructures development is unlikely to match the rate at which drone transportation and network connectivity are expected to penetrate the most remote places of the world. From a universal health coverage perspective, innovative interventions amenable to a multi-disease approach such as drones are the need of the hour. More

specifically within TB vertical programs, creative solutions like DrOTS and its components can support TB control and improve healthcare delivery in resource limited settings. Local regulations, costs and logistical challenges will determine the adoption of those technologies moving forward. This study found DrOTS to be a cost-effective intervention in remote Madagascar and suggests that such approaches may help finding some of the missing millions of TB cases in LMICs. This adds urgency to the need to overcome the regulatory, policy and logistical challenges that constrain the adoption of such technologies moving forward.

## Acknowledgments

We are grateful to the DrOTS project team including health teams involved in patient and data management. We also thank the Madagascar's National Tuberculosis Control Program personnel for their collaboration.

## Author Contributions

**Conceptualization:** Lulua Bahrainwala, James G. Kahn, Elizabeth Fair, Simon Grandjean Lapierre.

**Data curation:** Lulua Bahrainwala, Jesse McKinney, Simon Grandjean Lapierre.

**Formal analysis:** Lulua Bahrainwala, Astrid M. Knoblauch, Simon Grandjean Lapierre.

**Funding acquisition:** Peter M. Small, Niaina Rakotosamimanana.

**Investigation:** Lulua Bahrainwala, Andry Andriamiadanarivo, Jesse McKinney.

**Methodology:** Lulua Bahrainwala, Mohamed Mustafa Diab, James G. Kahn, Elizabeth Fair, Simon Grandjean Lapierre.

**Project administration:** Lulua Bahrainwala, Peter M. Small, Elizabeth Fair, Niaina Rakotosamimanana, Simon Grandjean Lapierre.

**Software:** Lulua Bahrainwala, Mohamed Mustafa Diab, James G. Kahn, Simon Grandjean Lapierre.

**Supervision:** Peter M. Small, Elizabeth Fair, Simon Grandjean Lapierre.

**Visualization:** Lulua Bahrainwala, Astrid M. Knoblauch, Simon Grandjean Lapierre.

**Writing – original draft:** Lulua Bahrainwala.

**Writing – review & editing:** Astrid M. Knoblauch, Andry Andriamiadanarivo, Mohamed Mustafa Diab, Jesse McKinney, Peter M. Small, James G. Kahn, Elizabeth Fair, Niaina Rakotosamimanana, Simon Grandjean Lapierre.

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
