## [Decision Letter · Decision Letter 0]

15 May 2020

PONE-D-20-10161

Drones and digital adherence monitoring for community-based tuberculosis control in remote Madagascar: a cost-effectiveness analysis

PLOS ONE

Dear Dr. Lapierre,

Thank you for submitting your manuscript to PLOS ONE. After careful consideration, we feel that it has merit but does not fully meet PLOS ONE’s publication criteria as it currently stands. Therefore, we invite you to submit a revised version of the manuscript that addresses the points raised during the review process.

We would appreciate receiving your revised manuscript. To enhance the reproducibility of your results, we recommend that if applicable you deposit your laboratory protocols in protocols.io, where a protocol can be assigned its own identifier (DOI) such that it can be cited independently in the future. For instructions see: http://journals.plos.org/plosone/s/submission-guidelines#loc-laboratory-protocols

We look forward to receiving your revised manuscript.

Kind regards,

Frederick Quinn

Academic Editor

PLOS ONE

Journal Requirements:

3. Thank you for stating the following in the Competing Interest section:

The authors have declared that no competing interests exist.  

We note that one or more of the authors are employed by a commercial company: ValBio Research Center.

Reviewers' comments:

Reviewer's Responses to Questions

**Comments to the Author**

1. Is the manuscript technically sound, and do the data support the conclusions?

Reviewer #1: Yes

Reviewer #2: Yes

2. Has the statistical analysis been performed appropriately and rigorously? 

Reviewer #1: Yes

Reviewer #2: Yes

3. Have the authors made all data underlying the findings in their manuscript fully available?

Reviewer #1: Yes

Reviewer #2: Yes

4. Is the manuscript presented in an intelligible fashion and written in standard English?

Reviewer #1: Yes

Reviewer #2: Yes

5. Review Comments to the Author

Reviewer #1: This manuscript compares ICER regarding conventional DOTS and innovative DrOTS in TB diagnosis, treatment and management, which would be helpful to improve the TB patients’ access to universal health care in resource-limited regions.

However, there are a few questions to be clarified from the author

1.How to deal with the retreatment TB patients, whose treatment course should be 8 months according to WHO guidelines?

2.The difficulties and challenged encountered during implementing DrOTS should be discussed in detail in order to help others to better understand the innovative intervention, such as how to make sure that the drones could deliver medication to the right patient?

3.It seems that TB patients under DrOTS will not visit outpatient clinic and see the doctor, so how to do when there is side effect and how to evaluate the treatment outcome for smear-negative without taking chest X-ray examination?

Reviewer #2: The manuscript is well written, comprehensive data analyses are done and clearly presented. The conclusion are meaningful. The findings of the study address a very important and novel area of research where evidence is still very limited. The findings are very informative and will be very useful for improving health systems in Sub-Saharan Africa.

6. PLOS authors have the option to publish the peer review history of their article (what does this mean?). If published, this will include your full peer review and any attached files.

Reviewer #1: No

Reviewer #2: Yes: Juliet Nabbuye Sekandi

---

## [Author Response · Author response to Decision Letter 0]

18 May 2020

Manuscript Number: PONE-D-20-10161

Manuscript Title: Drones and digital adherence monitoring for community-based tuberculosis control in remote Madagascar: a cost-effectiveness analysis

Journal: PLOS ONE

Montreal May 16th 

Dear Frederick Quinn, editor 

We wish to. thank you and the two external reviewers for reviewing our paper in detail and providing useful comments and suggestions. 

We have carefully reworked our manuscript in light of your reports. Below, please find our point-by-point response, clearly indicating how and where in the manuscript (line numbers) changes have been made. Altered text has been highlighted throughout the manuscript using a yellow marker to assist you in readily reviewing our changes made.

We feel that our revised manuscript has gained in clarity and quality, and hence, we hope that it is now suitable for publication in PLOS One. 

Yours sincerely,

Dr. Simon Grandjean Lapierre MD MSc FRCPC

On behalf of all the authors

 

EDITOR’S COMMENTS 

COMMENT #1 – Please ensure that your manuscript meets PLOS ONE's style requirements, including those for file naming. 

ANSWER #1 – The formatting was adjusted as requested in the reviewed version. (See highlights throughout the manuscript)

COMMENT #2 – We note that you have indicated that data from this study are available upon request. PLOS only allows data to be available upon request if there are legal or ethical restrictions on sharing data publicly. For information on unacceptable data access restrictions, please see http://journals.plos.org/plosone/s/data-availability#loc-unacceptable-data-access-restrictions.

AMSWER #2 – There are no ethical or legal restrictions on sharing this study’s data set. This is a modeling study. The primary data for this study is as described in the “Decision analysis model” and “Costs” sections and in “Table 1” and “Table 2”. It includes Madagascar’s National TB program data and TB disease pathophysiology information which is already in the public domain and appropriately referenced. It also includes. Madagascar’s TB clinics operation cost and multiple bills and financial statements from the DrOTS project and implementing institutions. We believe the methods and detailed data sources provided in the manuscript text allows for reproduction of our results. If further proof of data is needed (e.g. bill of drone purchase, bill of electricity for drone charging) we are committed to providing this information to whom would be interested but feel it is not a dataset which would be helpful to the scientific community on a public repository. Please do update our Data Availability statement on our behalf to reflect this information.

COMMENT #3 – Thank you for stating the following in the Competing Interest section: The authors have declared that no competing interests exist. 

ANSWER #3 – This was added to the manuscript. (Line 597)

COMMENT #4 – We note that one or more of the authors are employed by a commercial company: ValBio Research Center. Please provide an amended Funding Statement.

ANSWER #4 – ValBio Research Center is not a commercial company. It is a Research Campus owned by Stonybrook University and operating in Madagascar under a Non-Profit Organization legal status. It is part of the academic institution. ValBio Research is therefore not a sponsor or beneficiary of this study. Hence, we did not modify any of the authors’ contribution or Competing Interests Statement. Regarding the Funder (Stop TB Partnership), we have amended the Funding Statement as suggested. The Funding Statement and Competing Interests Statement should now read as follows. We thank the editorial team for making those changes on the online submission form on our behalf. (Line 605)

Competing interest 

The authors have declared that no competing interest exist.

Financial disclosure

This work was supported by the Stop TB Partnership’s TB REACH wave 5 award to NR and PMS (http://www.stoptb.org/global/awards/tbreach/wave5.asp). The Stop TB Partnership is funded by the Government of Canada and the Bill and Melinda Gates Foundation. AMK is supported by the Rudolf Geigy Foundation, Swiss Tropical and Public Health Institute, Basel, Switzerland. SGL is supported by the Canadian Association for Microbiology and Infectious Diseases. The funder did not provide salary support for principal investigators but did provide salary support for research staff including a co-author on this manuscript (AA). The specific roles of the authors are articulated in the ‘author contributions’ section.” No funders or sponsors played any role in study design, data collection and analysis, decision to publish or preparation of the manuscript.

REVIEWER 1 COMMENTS 

COMMENT #1 – How to deal with the retreatment TB patients, whose treatment course should be 8 months according to WHO guidelines?

ANSWER #1 – Indeed, in this modeling study, we focused on first episodes of drug-susceptible pulmonary TB. To adapt the DrOTS model to different case scenarios including re-treatment cases but also, pediatric TB, extrapulmonary TB and Drug-Resistant TB, would require the construction of different models which would be informed by additional primary collected data. We acknowledge this limitation in the discussion, and we have modified the text to better reflect the additional challenge raised by the reviewer. (Line 554)

COMMENT #2 – The difficulties and challenged encountered during implementing DrOTS should be discussed in detail in order to help others to better understand the innovative intervention, such as how to make sure that the drones could deliver medication to the right patient?

ANSWER #2 – We have provided additional details in the “Modeled strategies / DrOTS” sub-section of the manuscript. (Line 153)

COMMENT #3 – It seems that TB patients under DrOTS will not visit outpatient clinic and see the doctor, so how to do when there is side effect and how to evaluate the treatment outcome for smear-negative without taking chest X-ray examination?

ANSWER #3 – Treatment outcomes for smear-positive cases are assessed with control smears as described in the manuscript. In the decision analysis tree, enrollment in the DOTS or DrOTS arms is based on microbiology testing (either smear or GeneXpert). If not recognized upon first testing, patients are expected to seek medical attention again later in disease course and potentially be diagnosed at this time. This model does not account for smear-negative patients not being diagnosed upon two consecutive testing. As it is the case for pediatric TB, drug resistant TB or extrapulmonary TB, DrOTS does not currently allow for comprehensive management of all TB forms in a de-centralized approach and support from local health centers is still required for certain patients. Although those patients are equally accounted for in both modeled strategies, we do not believe that this impacts our conclusions in terms of ICER or absolute cost per additional TB patient diagnosed. 

REVIEWER 2 COMMENTS 

COMMENT #2 – The manuscript is well written, comprehensive data analyses are done and clearly presented. The conclusion are meaningful. The findings of the study address a very important and novel area of research where evidence is still very limited. The findings are very informative and will be very useful for improving health systems in Sub-Saharan Africa.

ANSWER #2 – We thank reviewer 2 for seeing the added value of our study to the currently available literature.

---

## [Decision Letter · Decision Letter 1]

4 Jun 2020

PONE-D-20-10161R1

Drones and digital adherence monitoring for community-based tuberculosis control in remote Madagascar: a cost-effectiveness analysis

PLOS ONE

Dear Dr. Dr. Lapierre,

Thank you for submitting your manuscript to PLOS ONE. After careful consideration, we feel that it has merit but does not fully meet PLOS ONE’s publication criteria as it currently stands. Therefore, we invite you to submit a revised version of the manuscript that addresses the points raised during the review process.

Please submit your revised manuscript. If you will need more time than this to complete your revisions, please reply to this message or contact the journal office at plosone@plos.org. Please include the following items when submitting your revised manuscript:

We look forward to receiving your revised manuscript.

Kind regards,

Frederick Quinn

Academic Editor

PLOS ONE

Reviewers' comments:

Reviewer's Responses to Questions

**Comments to the Author**

1. If the authors have adequately addressed your comments raised in a previous round of review and you feel that this manuscript is now acceptable for publication, you may indicate that here to bypass the “Comments to the Author” section, enter your conflict of interest statement in the “Confidential to Editor” section, and submit your "Accept" recommendation.

Reviewer #1: All comments have been addressed

Reviewer #2: All comments have been addressed

Reviewer #3: All comments have been addressed

2. Is the manuscript technically sound, and do the data support the conclusions?

Reviewer #1: Yes

Reviewer #2: Yes

Reviewer #3: Yes

3. Has the statistical analysis been performed appropriately and rigorously? 

Reviewer #1: Yes

Reviewer #2: Yes

Reviewer #3: N/A

4. Have the authors made all data underlying the findings in their manuscript fully available?

Reviewer #1: Yes

Reviewer #2: Yes

Reviewer #3: Yes

5. Is the manuscript presented in an intelligible fashion and written in standard English?

Reviewer #1: Yes

Reviewer #2: Yes

Reviewer #3: Yes

6. Review Comments to the Author

Reviewer #1: The authors have responsed to the comments point by point and everything is clear. It's worth publishing.

Reviewer #2: The authors have duly addressed all comments raised by reviewers and there are no further comments. The manuscript is well done.

Reviewer #3: Title: The title is clear and conveys the essence of the authors’ work.

Background: The background and literature is exhaustive and introduces the main research idea adequately.

93: the use of “and” is not unnecessary in this sentence.

Methods: The aspect of the temporality of the DOTs and DrOTS study is not well elaborated. Did both aspects occur at the same time? It is important for you to clarify whether participants were studied at the same time and for what period.

148: In your methods, you make general statements in present tenses. For instance, you do not have to generally inform your readers about DOTS or other methods; references will suffice. Otherwise re-write the methods placing these in context of what you did, preferably in the past tense.

114: Could you convey your study data collection method into one of the conventional epidemiological designs, such as cross-sectional/survey/observational design etc

118: Did you seek participant consent and assent from minors? If not please clarify if a waiver to that effect was granted.

123: is Ifanadiana in Southeastern part or south-central as indicated inline 91?

181: Typo, “to monitor”

212: Is Malagasy districts a typo?

255: Any reason why drug-resistant TB was not accounted for in DALY The methods section is comprehensive and clear but you present a lot of information in the text. I suggest that you present your model inputs and costing in including references in tables. This will reduce the word count.

See my comments below regarding sensitivity analysis.

Results:

398: Your result that shows that of the 500 prevalent TB cases, 256 (51.3%) and 95 (19%) deaths were recorded is unbelievably high!

399: By you saying, “most of this difference was due to case-finding and treatment initiation”. You are presenting a discussion to your results that is non-contextual. Please clarify or move this statement to the discussion.

As a way to reduce word count, I suggest that the authors move the sensitivity analysis results to supplementary material.

Discussion: The first paragraph fits in the background better and paragraph 2 can serve as your first here.

523: Typo, “ very cost-effectivene threshold”

534: “This study is the first study to present a cost-effectiveness evaluation…”, maybe you say, “To the best of our knowledge…” otherwise your assertion needs a reference.

The limitations of your CEA model are well written!

7. PLOS authors have the option to publish the peer review history of their article (what does this mean?). If published, this will include your full peer review and any attached files.

Reviewer #1: Yes: Fei Huang

Reviewer #2: Yes: Juliet N. Sekandi

Reviewer #3: No

---

## [Author Response · Author response to Decision Letter 1]

10 Jun 2020

Montreal June 4th 

Dear Frederick Quinn, editor 

We wish to thank you and the three external reviewers for reviewing our paper in detail and providing useful comments and suggestions. 

We are pleased to see that the reviewers #1 & #2 were entirely satisfied with provided revisions. 

We are now receiving comments from reviewer #3 for the first time and are happy to provide additional precisions as requested. We realize that reviewer #3 provides line references to the initial submitted manuscript rather that the subsequent revised version. We also realise that many comments point to a conceptual misunderstanding of this study which is a modelled cost-effectiveness study rather than an interventional trial. Hence issues such as informed consent or study temporality are confusing. This is addressed in detail in our point-by-point response below. We clearly indicate how and where in the manuscript (line numbers) changes have been made. Altered text has been highlighted throughout the manuscript using a yellow marker to assist you in readily reviewing our changes made.

We feel that our revised manuscript has gained in clarity and quality, and hence, we hope that it is now suitable for publication in PLOS One. 

Yours sincerely,

Dr. Simon Grandjean Lapierre MD MSc FRCPC

On behalf of all the authors

Reviewer #1 comments

COMMENT #1 - The authors have responded to the comments point by point and everything is clear. It's worth publishing.

ANSWER #1 – Well noted. 

Reviewer #2 comments 

COMMENT #2 - The authors have duly addressed all comments raised by reviewers and there are no further comments. The manuscript is well done.

ANSWER #2 – Well noted. 

Reviewer #3 comments 

COMMENT #1 - The title is clear and conveys the essence of the authors’ work.

ANSWER #1 – Well noted 

COMMENT #2 - Background: The background and literature is exhaustive and introduces the main research idea adequately.

ANSWER #2 – Well noted. 

COMMENT #3 -93: the use of “and” is not unnecessary in this sentence.

ANSWER #3 – This was modified in the revised manuscript. (Line 89)

COMMENT #4 - Methods: The aspect of the temporality of the DOTs and DrOTS study is not well elaborated. Did both aspects occur at the same time? It is important for you to clarify whether participants were studied at the same time and for what period.

ANSWER #4 – This is a modeled cost-effectiveness study. It models costs, TB outcomes, DALYs and CE based on field collected and literature data. In terms of the temporality of the model input data collection, data for both DrOTS and DOTS were collected at the same time. This is now clearly indicated in the methods (Line 140). 

COMMENT #5 - 148: In your methods, you make general statements in present tenses. For instance, you do not have to generally inform your readers about DOTS or other methods; references will suffice. Otherwise re-write the methods placing these in context of what you did, preferably in the past tense.

ANSWER #5 – Given the modelling nature of this study, we feel the readership needs to be presented with a clear description of both modelled strategies. We also feel that using the present tense to describe what exactly are DOTS and DrOTS is appropriate. 

COMMENT #6 - 114: Could you convey your study data collection method into one of the conventional epidemiological designs, such as cross-sectional/survey/observational design etc

ANSWER #6 – This is a modelled Cost-Effectiveness study. Those conventional epidemiological designs do not apply to the current study. To increase clarity, we clearly stated this in the beginning of the methods section (Line 112). 

COMMENT #7 - 118: Did you seek participant consent and assent from minors? If not please clarify if a waiver to that effect was granted.

ANSWER #7 - This is a modelled Cost-Effectiveness study. This is not the report of an interventional trial. Informed consents are not required for modelling studies as those are based on field collected data but do not incur any intervention at patient level. 

COMMENT #8 - 123: is Ifanadiana in Southeastern part or south-central as indicated inline 91?

ANSWER #8 – Southeastern. This was adjusted in the manuscript (Line 88). 

COMMENT #9 - 181: Typo, “to monitor”

ANSWER #9 – This was adjusted in the revised manuscript (Line 183). 

COMMENT #10 - 212: Is Malagasy districts a typo?

ANSWER #10 – No 

COMMENT #11 - 255: Any reason why drug-resistant TB was not accounted for in DALY 

ANSWER #11 - Costs and DALYs associated with hospitalization, longer treatment course and drug adverse events management in MDR-TB treatment are excluded since this represents less than 1% of cases in Madagascar and has never been diagnosed in the study area. This is why it is not included in the model. This is included in the manuscript (Line 340).

COMMENT #12 - The methods section is comprehensive and clear but you present a lot of information in the text. I suggest that you present your model inputs and costing in including references in tables. This will reduce the word count.

ANSWER #12 – Previous comments we had to address from the other reviewers led to adding information in the methods sections because some elements were not clear enough. In this context, we decide to leave the methods section as is in order to ensure the readership can refer back to the text for detailed information on the costing (and other) model assumptions. 

COMMENT #13

398: Your result that shows that of the 500 prevalent TB cases, 256 (51.3%) and 95 (19%) deaths were recorded is unbelievably high

ANSWER #13 – This indeed what the model suggests based on all the provided, detailed and referenced assumptions. 

COMMENT #14 - 399: By you saying, “most of this difference was due to case-finding and treatment initiation”. You are presenting a discussion to your results that is non-contextual. Please clarify or move this statement to the discussion.

ANSWER #14 – Those are the. nodes of the decision tree model which have the most impact on mortality. These are findings of the model rather than hypotheses or elements of context. To clarify this, we have modified the statement in the revised manuscript (Line 401). 

COMMENT #15 - As a way to reduce word count, I suggest that the authors move the sensitivity analysis results to supplementary material.

ANSWER #15 – We believe the sensitivity analysis provides key information regarding this study’s external validity and potential for DrOTS implementation and impact in other high TB burden settings. Given the manuscript length and word count was not raised by the editor or the two other reviewers, we suggest keeping this in the original manuscript. 

COMMENT #16 - Discussion: The first paragraph fits in the background better and paragraph 2 can serve as your first here.

ANSWER #16 – Those changes were made to the manuscript as suggested (Line 76). 

COMMENT #17 -523: Typo, “ very cost-effectivene threshold”

ANSWER #17 – This was adjusted in the revised manuscript (Line 516). 

COMMENT #18 - 534: “This study is the first study to present a cost-effectiveness evaluation…”, maybe you say, “To the best of our knowledge…” otherwise your assertion needs a reference.

ANSWER #18 – This was adjusted in the revised manuscript (Line 526). 

COMMENT #19 - The limitations of your CEA model are well written!

ANSWER #19 – Well noted.

---

## [Decision Letter · Decision Letter 2]

18 Jun 2020

Drones and digital adherence monitoring for community-based tuberculosis control in remote Madagascar: a cost-effectiveness analysis

PONE-D-20-10161R2

Dear Dr. Lapierre,

We’re pleased to inform you that your manuscript has been judged scientifically suitable for publication and will be formally accepted for publication once it meets all outstanding technical requirements.

Kind regards,

Frederick Quinn

Academic Editor

PLOS ONE

Additional Editor Comments (optional):

Reviewers' comments:

Reviewer's Responses to Questions

**Comments to the Author**

1. If the authors have adequately addressed your comments raised in a previous round of review and you feel that this manuscript is now acceptable for publication, you may indicate that here to bypass the “Comments to the Author” section, enter your conflict of interest statement in the “Confidential to Editor” section, and submit your "Accept" recommendation.

Reviewer #1: All comments have been addressed

Reviewer #2: All comments have been addressed

2. Is the manuscript technically sound, and do the data support the conclusions?

Reviewer #1: Yes

Reviewer #2: Yes

3. Has the statistical analysis been performed appropriately and rigorously? 

Reviewer #1: Yes

Reviewer #2: Yes

4. Have the authors made all data underlying the findings in their manuscript fully available?

Reviewer #1: Yes

Reviewer #2: Yes

5. Is the manuscript presented in an intelligible fashion and written in standard English?

Reviewer #1: Yes

Reviewer #2: Yes

6. Review Comments to the Author

Reviewer #1: the authors have adjusted the manuscript based on the comments point by point and no more need to be clarified.

Reviewer #2: The authors have addressed all reviewers comments satisfactorily and i recommend to go forward with the next steps.

7. PLOS authors have the option to publish the peer review history of their article (what does this mean?). If published, this will include your full peer review and any attached files.

Reviewer #1: Yes: Fei Huang

Reviewer #2: Yes: Juliet N. Sekandi

---

## [Editor Report · Acceptance letter]

24 Jun 2020

PONE-D-20-10161R2 

Drones and digital adherence monitoring for community-based tuberculosis control in remote Madagascar: a cost-effectiveness analysis 

Dear Dr. Grandjean Lapierre:

I'm pleased to inform you that your manuscript has been deemed suitable for publication in PLOS ONE. Congratulations! Your manuscript is now with our production department. 

Kind regards, 

on behalf of

Dr. Frederick Quinn 

Academic Editor

PLOS ONE